# Changes in context, typology and programme outcomes between early and recent periods of sex work among young female sex workers in Mombasa, Kenya: A cross-sectional study

**Parinita Bhattacharjee**[1,2]☉\*, **Shajy Isac**[1]☉, **Helgar Musyoki**[3], **Peter Gichangi**[4], **Huiting Ma**[5], **Marissa Becker**[1], **Jan Hontelez**[6,7], **Sharmistha Mishra**[5,8,9], on behalf of the Transitions team¶

**1** Institute for Global Public Health, University of Manitoba, Winnipeg, Canada, **2** Partners for Health and Development in Africa, Nairobi, Kenya, **3** National AIDS and STI Control Programme, Ministry of Health, Nairobi, Kenya, **4** International Centre for Reproductive Health- Kenya, Mombasa, Kenya, **5** St. Michael's Hospital, Unity Health Toronto, Toronto, Canada, **6** Department of Public Health, Erasmus MC, Rotterdam, Netherlands, **7** Heidelberg Institute of Global Health, Heidelberg University, Heidelberg, Germany, **8** Institute of Medical Sciences, University of Toronto, Toronto, Canada, **9** Institute of Health Policy Management and Evaluation, Dalla Lana School of Public Health, University of Toronto, Toronto, Canada

☉ These authors contributed equally to this work.
¶ Complete membership of the author group can be found in the Acknowledgments.
\* Bhattacharjee.parinita@gmail.com

## Abstract

### Introduction

The sex work context and typology change continuously and influence HIV related risk and vulnerability for young female sex workers (YFSW). We sought to describe changes in the context and typology of sex work between the first (early) and past month (recent) of sex work among YFSW to inform HIV prevention programming for sex workers.

### Methods

We used data from a cross-sectional survey (April-November 2015), administered using physical location-based sampling to 408 cis-women, aged 14–24 years, who self-identified as sex workers, in Mombasa, Kenya. We collected self-reported data on the early and recent month of sex work. The analysis focused on changes in a) sex work context and typology (defined by setting where sex workers practice sex work) where YFSW operated, b) primary typology of sex work, and c) HIV programme outcomes among YFSW who changed primary typology, within the early and recent month of sex work. We analysed the data using a) SPSS27.0 and excel; b) bivariate analysis and χ2 test; and c) bivariate logistic regression models.

### Results

Overall, the median age of respondents was 20 years and median duration in sex work was 2 years. Higher proportion of respondents in the recent period managed their clients on their

**Data Availability Statement:** The data contain information related to locations where young

women who sell sex conduct sex work. Sex work and sex workers are criminalised in Kenya, hence the data is sensitive. Data are available on request from the IGPH - University of Manitoba Data Manager, Stella Leung, Stella.Leung@umanitoba.ca.

**Funding:** The study was funded by an operating grant (MOP-13044) from the Canadian Institutes of Health Research (CIHR) and analyses funded via CIHR grant FDN 13455. MB, SM were supported by CIHR grants - MOP 13044 and FDN13455. PB was supported by BMGF grant - INV006613. The funders had no role in study design, data collection and analysis, decision to publish or preparation of the manuscript. There was no additional external funding received for this study.

**Competing interests:** The authors have declared that no competing interests exist.

**Abbreviations:** YFSW, Young Female Sex Worker; FSW, Female Sex Workers; STI, Sexually Transmitted Infections; NGO, Non-Governmental Organisation; FBO, Faith based Organisation; CBO, Community based Organisation; OR, Odds Ratio; CI, Confidence Interval.

own (98.0% vs. 91.2%), had sex with >5 clients per week (39.3% vs.16.5%); were able to meet > 50% of living expenses through sex work income (46.8% vs. 18.8%); and experienced police violence in the past month (16.4% vs. 6.5%). YFSW reported multiple sex work typology in early and recent periods. Overall, 37.2% reported changing their primary typology. A higher proportion among those who used street/ bus stop typology, experienced police violence, or initiated sex work after 19 years of age in the early period reported a change. There was no difference in HIV programme outcomes among YFSW who changed typology vs. those who did not.

## Conclusions

The sex work context changes even in a short duration of two years. Hence, understanding these changes in the early period of sex work can allow for development of tailored strategies that are responsive to the specific needs and vulnerabilities of YFSW.

## Introduction

Limited information on the context and settings of sex work during the early period of sex work poses a significant challenge for HIV prevention programmes to design and adapt services to reach women at the time they enter sex work [1]. There are several studies that show that the setting in which women practice sex work, and context in which sex work takes place, changes over time [2–4] but very few studies have attempted to understand these changes [5]. The importance of understanding the context and organization of the sex work industry [6–8] and different settings [9] particularly during the early stages of sex work [10] is gaining increasing attention from HIV prevention programmes to ensure that delays in HIV programme contact for young female sex workers (YFSW) 15–25 years, are reduced and interventions strategies are developed to effectively address their specific needs and vulnerabilities.

The ways in which sex work is practiced are diverse and change over time responsive to sociocultural, economic or technological context [11, 12]. Both context and setting of sex work affect the health and personal safety of sex workers, as identified by Harcott et al [13]. One of the critical changes in the setting in which sex workers work can be understood and described as the typology of sex work. Typology of sex work is defined as the classification of female sex workers (FSW) into types, or categories, based on the settings in which they practice sex work such as street setting or lodge setting [14]. Understanding typology is important as sex workers in different settings experience a range of unique risks and vulnerabilities and therefore require targeted programming strategies [15–18]. Several studies across countries have found that HIV and other sexually transmitted infections (STI) prevalence among FSWs [19, 20] and the reach of the HIV prevention programmes [21], differ by sex work typologies. Differences in the context and working conditions in varied sex work typology can influence sex work practices and potential risks of HIV infection [22]. There can be different clusters of risk factors associated with different typology of sex work, thus demanding prevention programmes to tailor interventions based on typology of sex work [23]. Sex work typologies are not static as FSWs constantly change their settings. Mishra et al. in their study in Nepal found that the changing typology of sex work increases the vulnerability of sex workers to STI and HIV infections as these changes result in the sex workers getting cutoff from HIV programmes due to low awareness of programmes in the new locations, posing challenges for them to access

information and services [24]. Understanding the diversity in sex work typology and the context of sex work is crucial for assessing the risk of acquiring and transmitting HIV and other STIs among FSW [25, 26]. It also has important implications for planning and implementing effective HIV and STI interventions that are tailored to the specific needs of this population [8, 9, 16, 27–29].

YFSW are at particularly heightened risk of HIV in Africa [30]. In addition to the physiological, emotional and social vulnerabilities faced by adolescent girls and young women as they transition into adulthood, YFSW face added challenges related to stigma, discrimination and criminalisation [31, 32], and reduced ability to negotiate condom use with sexual partners [4]. Most HIV prevention programmes tailored for FSWs are designed to reach women several years after entering sex work or self-identifying as sex worker [33]. It is important to understand the changes in the context and typology of sex work during the early period of sex work in order to design effective time–period tailored HIV prevention interventions for YFSW who are just beginning sex work.

In this paper we sought to describe the changes in context and typology of sex work and programme outcomes during the first (early) and past (recent) month of sex work among YFSW in Mombasa, Kenya. We describe the changes in a) context of sex work; b) distribution of typologies of sex work including extent and patterns of overlap; and c) primary typology of sex work within person, during the early and recent periods of sex work. Then we describe the characteristics of YFSW during the early period of sex work for those sex workers who change their primary typology. In addition, we compare and describe the programme outcomes specifically prevalent HIV infection, HIV programme reach and service utilization among those YFSW who change their primary typology during the early and recent period of sex work vs. those who do not.

## Methods

We used data from a bio-behavioural cross-sectional survey, administered between April and November 2015, to sexually active cis gender adolescent girls and young women aged 14–24 years frequenting sex work physical locations in Mombasa, Kenya. Sex work locations included any locale or setting where individuals solicited clients for sex or provided sex services in exchange for money [34]. Eligibility criteria for the survey included: sexually active females, aged 14–24 years, and provision of written informed consent. Consent was obtained from participants directly to participate in the study. Written informed consent from the participants' legal guardian/next of kin was not required for participation in this study in accordance with the national legislation and the institutional requirements [35]. For the paper, we restricted our analyses to data on 408 participants who self-identified as sex workers.

We identified sex work locations using geographic mapping [36], which also provided the distribution and population size of YFSW. The list of sex work locations, served as the sampling frame, and a sample of 85 locations were selected via probability proportional to the estimated YFSW population size at each location. Peer/community researchers (i.e. current or former sex workers engaged with International Centre for Reproductive Health–Kenya (ICRH-K)), a local HIV prevention organisation, went to the sampled locations and selected potential participants for eligibility screening and consent as detailed previously [37]. Trained interviewers used structured questionnaires to conduct face-to-face interviews in English or Kiswahili. Details of the data collection have been detailed previously [38].

The participants were reimbursed Ksh 850 for their time. The survey took 1 hour and 30 minutes including the collection of biological samples. The survey took place in private spaces that were located within the Drop-in Centres and clinics managed by ICRH-K or government clinics identified by the peer educators as safe and friendly spaces. All the rooms where the

face-to-face interviews and biological sample collections took place had privacy for the respondents. All respondents were given a unique ID to protect confidentiality and this ID was attached to their questionnaire and biological sample. Hard copies of the filled questionnaires and other documents were stored in locked cabinets, accessible only by members of the research team. The electronic data was entered in a password protected computer and stored in a password protected server. Only the research team had access to the data.

## Measurement and data analysis

In this paper, we define **early** period of sex work as the first month of sex work practice after the respondent self-identified herself as sex worker and **recent** period of sex work as the last month of sex work practice before the respondent participated in the survey. To understand the changes in sex work practices in the early and recent period of sex work, we used the following variables, a) management of sex work, b) number of paying clients per week, c) income from sex work to meet living expenses, d) experience of coercion in the past month and e) experience of police violence in the past month. Typology refers to the classification of YFSW based on settings in which sex work took place (either solicitation and/or site of the sexual service). We used the following question; *how did you meet your paying clients in the first month of sex work and in the last month of sex work* to define typology. The question was a multiple answer question with 15 response options. We combined the options to define five typologies of sex work, namely a) internet and mobile phone b) street and bus stop c) entertainment venue d) hotel and lodge and e) others. The others typology included places like home, school, church etc. We compared the typology distribution in the early to the recent period to identify any changes that may have occurred. Furthermore, we assessed the extent and pattern of overlap in the typology of sex work among the YFSWs during these two periods. To define primary typology in the paper, we used the question, *of all the different places and ways you had met paying clients, which was the most common for the first month of sex work and the last month of sex work*. We assessed within person change in primary typology between the early and recent period of sex work. Among YFSW who reported change in the primary typology of sex work, we examined their following characteristics during the early period of sex work: a) primary typology of sex work; b) management of sex work; c) meeting living expenses through income of sex work; d) number of paying clients per week; e) experience of physical violence in the past month; f) experience of coercion in the past month; g) experience of police violence in the past month; h) living condition and i) age at start of sex work. We then assessed the change in programme outcomes i.e. prevalent HIV infection, HIV programme reach and service utilisation among YFSW who changed their primary typology compared to those who did not. We used HIV test result (conducted on site) to assess the prevalent HIV infections. We used one questions to measure programme reach i.e. ever contacted by a peer or staff from Non-Governmental Organization (NGO)/ Community based Organisation (CBO)/ Faith based Organisation (FBO); and two questions to measure service utilisation namely a) ever tested for HIV and; b) ever used a clinic run by NGO/ CBO/ FBO.

Details of the questions used to define each variable is presented as S1 Table.

## Statistical analysis

The importance and value of descriptive work in epidemiology and public health is well established. Descriptive studies seek to characterize what is happening in the world to inform public health priorities and target interventions [39, 40]. We used descriptive statistics to describe the change in practice and context of sex work, change in typology distribution, extent and pattern of overlap in typology of sex work; and change in primary typology of sex work in early and

recent period of sex work for YFSW. We also conducted sensitivity analysis to assess the change in primary typology of sex work by duration in sex work. We described the early period characteristics of those YFSW who changed their primary typology using bivariate analysis and χ2 test to assess the factors associated with the change in typology. We then assessed the programme outcomes (prevalent HIV infection, programme reach and service utilisation) for those FSW who changed their primary typology and compared with those who did not, using bivariate logistic regression models. We used SPSS27.0 and excel for statistical analyses and graphics and presented Odds Ratio (OR) and 95% Confidence Interval (CI).

### Inclusivity in global research

Additional information regarding the ethical, cultural, and scientific considerations specific to inclusivity in global research is included in S1 File.

### Ethics approval

The study received ethics approval from the Health Research Ethics Board at the University of Manitoba, Canada (HS16557 (H2013:295)) on 9th September 2014 and the Kenyatta National Hospital-University of Nairobi Ethics and Research Committee, Kenya (P497/10/2013) on 5th June 2014. The study also received a research permit from the National Commission for Science, Technology and Innovation (NACOSTI), Kenya.

## Results

Overall (Table 1), 42.2% of the YFSW respondents were less than 20 years with the median age being 20 years (IQR- 4), 69.6% had completed primary education, 16.5% had regular source of income, 58.1% respondents had their sexual debut between 15–19 years with the median age at sexual debut being 15 years (IQR– 3), 24.0% of the respondents were in sex work for less than one year and 40% were in sex work for 1–2 years, with the median duration being 2 years (IQR– 2).

### Change in sex work practices, context and typology

Table 2 compares the sex work practices among YFSW in the early and recent period of sex work. Compared to the early period, at the recent period, a higher proportion of YFSW managed solicitation of clients independently without the use of a third party (pimp or manager) (91.2% vs 98.0%, p < 0.001), had more than 5 paying clients per week (16.5% vs 39.3%, p < 0.001), had higher mean number of paying clients per week (4.0 clients vs 5.7 clients, p <0.001), met more than half of their living expenses through the income of sex work (18.8% vs 46.8%, p < 0.001) and had experienced police violence in the past month (6.5% vs 16.4%, p <0.001). However, there was no change reported in experienced coercion in the past month among YFSW (7.6% vs 4.6%, p = 0.180).

Table 3 shows the change in typology from the early to recent period of sex work with an increase in the proportion of YFSW reporting using internet and mobile phone (4.9% to 19.1%, p<0.001). However, no major changes were reported in the use of street and bus stops (45.6% to 40.2%, p = 0.119); entertainment venues (70.6% to 75.5%, p = 0.115); hotel and lodges (21.6% to 25.7%, p = 0.168), and other (40.7% to 35.8%, p = 0.150) typology of sex work between early and recent period.

### Change in extent and pattern of overlap of typology of sex work

Fig 1 (panel A and B) shows that at both the early and recent periods of sex work, YFSW used more than one sex work typology. In the early period of sex work, 100% of the internet and mobile phone based, 80.1% of street and bus stop based, 64.7% of entertainment based, 88.6%

**Table 1. Socio-demographic and sexual characteristics of young female sex workers who participated in the Transition study in Mombasa, Kenya, 2015.**

| Characteristics | Percent N = 408 |
|---|:---:|
| **Age** | |
| <20 years | 42.2 |
| 20–24 years | 57.8 |
| Median age (years) | 20.0 (IQR[‡]-4) |
| **Education level** | |
| Upto Primary | 30.4 |
| Primary complete | 31.1 |
| Attended high school and above | 38.5 |
| **Income** | |
| % currently have regular source of income[§] | 16.5 |
| **Age at first sex** | |
| <15 years | 34.8 |
| 15–19 years | 58.1 |
| 20–24 years | 7.1 |
| Median age at sexual debut | 15 years (IQR[‡]-3) |
| **Duration in sex work** | |
| <1 years | 24% |
| 1–2 years | 40% |
| 3–4 years | 23% |
| 5 years and more | 13% |
| Mean duration in sex work | 2.2 years (SD[†] - 2.2) |
| Median duration in sex work | 2 years (IQR[‡] - 2) |

[†]SD: Standard deviation

[‡]IQR: Interquartile range

[§] The questionnaire did not clearly define the source of income so we cannot say whether source of income includes income from sex work or not

hotel and lodge based and 80.1% YFSW practicing in other typology of sex work also practiced in more than 2 other typologies. Similarly, at the recent period of sex work, 94.7% of the internet and mobile phone based, 81.1% of street and bus stop based, 71.8% of entertainment based, 71.5% hotel and lodge based and 71.5% FSW practicing in other typology of sex work also practiced in more than 2 other typologies.

Fig 1 (panel C and D) shows that in the early and recent periods of sex work, most of the overlap in typologies were in entertainment venues. In the early period of sex work, 55.0% of YFSW from internet and phone-based typology, 65.1% from the street and bus stop typology, 76.1% from hotel and lodge typology and 59.6% from other typology also used entertainment venues. However, in the recent period, there was an increase in the overlap with entertainment venues. 85.9% of YFSW from internet and phone-based typology, 69.5% from the street and bus stop typology, 81.9% from hotel and lodge typology and 64.4% from other typology also used entertainment venues. The overall pattern of overlap in typologies changed between the early and recent period of sex work, with higher overlap with entertainment venues during the recent period of sex work.

### Pattern of within person change in primary typology of sex work

Overall, 37.2% (150/403) of the respondents reported changing their primary typology of sex work between their early and recent period of sex work. There was not much variation in

**Table 2. Change in practice of sex work and context in the early and recent period of sex work among young female sex workers in Mombasa, Kenya, 2015.**

| | | Early (N = 408) % | Recent (N = 408) % | P value |
|---|---|---|---|---|
| Management of sex work | Pimp | 6.4 | 1.5 | <0.001 |
| | Manager | 4.4 | 1.5 | |
| | Self-managed | 91.2 | 98.0 | |
| Number of paying clients per week | 0–2 | 46.2 | 39.3 | <0.001 |
| | 3–4 | 34.1 | 19.5 | |
| | 5 or more | 16.5 | 39.3 | |
| | Do not know/ did not answer | 3.2 | 1.9 | |
| | Mean number of paying clients per week (#) | 4.0 client (SD[†]-8.3) | 5.7 client (SD[†]-6.9) | <0.001 |
| Meet living expenses through income of sex work | Less than half | 30.2 | 15.0 | <0.001 |
| | Half | 51.0 | 38.2 | |
| | More than half | 18.8 | 46.8 | |
| Experienced coercion in the past month | | 7.6 | 4.6 | 0.18 |
| Experienced police violence in the past month | | 6.5 | 16.4 | <0.001 |

†SD: Standard deviation

change in primary typology of sex work by duration in sex work (S2 Table). Fig 2 shares within person change in primary typology between the early and recent period of sex work. Among those who changed primary typology, in the early period, 1.3% of the respondents reported being internet and mobile phone based, 38.0% street and bus stop based, 36.0% entertainment venue based, 5.3% hotel and lodge based and 19.3% reported other typologies of sex work. A shift was noticed among YFSW in all typologies as 80.7% of those street/bus stop based in the early period shifted to entertainment venues and another 14.0% shifted to other typologies. Among YFSW in entertainment venues in the early period, 24.1% shifted to internet/mobile phone, 29.6% shifted to street/bus stop, 18.5% shifted to hotel/lodge and the remaining 27.8% shifted to other typology of sex work. Similarly, from the hotel/lodge-based sex worker, 37.5% shifted to entertainment venues and the remaining 62.5% shifted to other typology of sex work. Among those in other typologies in the first month, a very large proportion shifted to entertainment venue (82.8%), and very few to street/bus stop (6.9%) and hotel/lodges (6.9%). These changes therefore contributed to a change in primary typology in the recent period as 10.0%, 12.7%, 49.3%, 8.0% and 20.0% reported their primary typology as internet and mobile phone based, street/bus stop based, entertainment venue based, hotel/lodge based and other typologies respectively.

**Table 3. Distribution in typology in early and recent period of sex work among young female sex workers, Mombasa, Kenya, 2015.**

| Typology of sex work[†] | Early N = 408 | Recent N = 408 | P Value |
|---|---|---|---|
| Internet/Mobile phone | 4.9% | 19.1% | 0.001 |
| Street/Bus stop | 45.6% | 40.2% | 0.119 |
| Entertainment venue | 70.6% | 75.5% | 0.115 |
| Hotel/Lodge | 21.6% | 25.7% | 0.168 |
| Others | 40.7% | 35.8% | 0.150 |

†Typology of sex work here is defined as multiple options and all the different typologies solicited in the respective period.

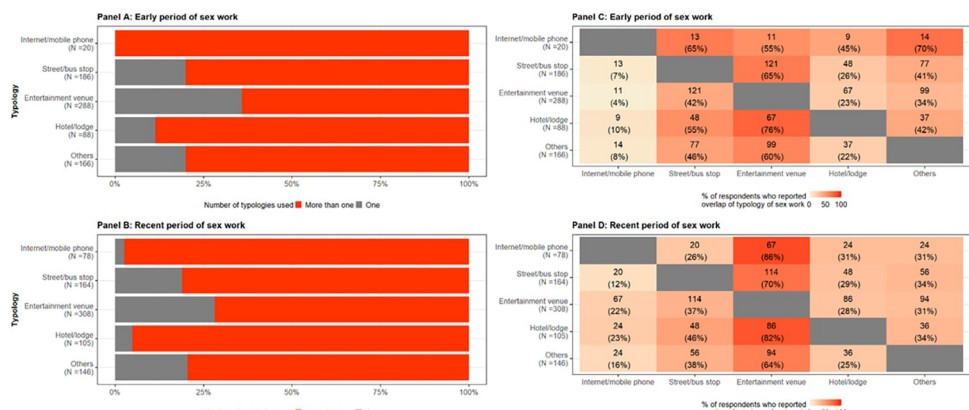

**Fig 1. Change in extent and pattern of overlap of typology of sex work between the early and recent period of sex work among young female sex workers in Mombasa, Kenya, 2015.**

## Characteristics of YFSW in the early period among those who changed their primary typology

Table 4 shows the characteristics in the early period of sex work for YFSW who changed their primary typology. Compared to those who did not change their primary typology, a higher proportion of YFSW who changed their primary typology were street and bus stop based (18.2% vs. 38.0%, p<0.001), experienced police violence in the past month (4.7% vs. 9.7%, p = 0.05) and initiated sex work after 19 years of age (33.3% vs. 45.3%, p<0.02).

## Change in programme outcomes among YFSW who changed their primary typology

In terms of programme outcomes, Table 5 shows that overall 14.7% YFSWs were ever contacted by a peer/staff from NGO/CBO/FBO; 77.4% ever tested for HIV; and 13.7% ever visited

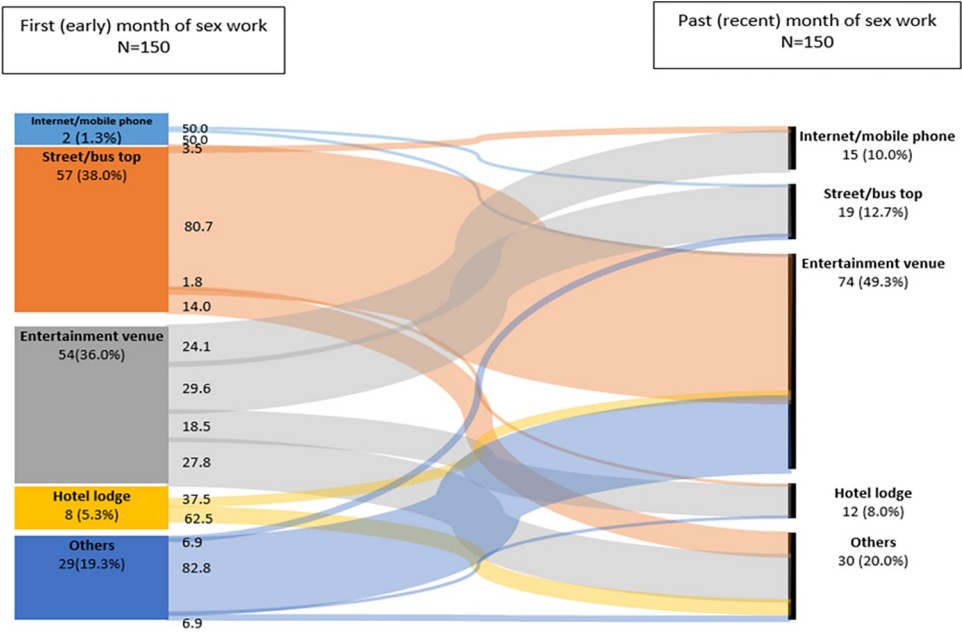

**Fig 2. Pattern of within person change in primary typology of sex work between the early and recent period of sex work among young female sex workers who changed their primary typology in Mombasa, Kenya, 2015.**

**Table 4. Characteristics of young female sex workers at early period of sex work, who changed their primary typology of sex work between early and recent period of sex work, Mombasa, Kenya, 2015.**

| | | First month of sex work | | | |
| | | Total | Change of typology | | P Value |
| | | | Changed | Not Changed | |
| | | N = 408 | n = 150 | n = 258 | |
|---|---|---|---|---|---|
| Primary typology | Internet/mobile phone | 0.5 | 1.3 | 0.0 | <0.001 |
| | Street/bus top | 25.5 | 38.0 | 18.2 | |
| | Entertainment venue | 53.7 | 36.0 | 64.0 | |
| | Hotel lodge | 4.9 | 5.3 | 4.7 | |
| | Others | 15.4 | 19.3 | 13.2 | |
| Management of sex work | Pimp | 6.4 | 8.0 | 5.4 | 0.27 |
| | Manager | 4.4 | 6.0 | 3.5 | |
| | Self | 91.2 | 90.0 | 91.8 | |
| Meet living expenses through income of sex work | Less than half | 30.2 | 35.1 | 27.3 | 0.21 |
| | Half | 51.0 | 45.9 | 53.9 | |
| | More than half | 18.8 | 18.9 | 18.8 | |
| Number of paying clients per week[†] | <3 | 47.7 | 50.7 | 46.0 | 0.60 |
| | 3–4 | 35.2 | 32.1 | 36.9 | |
| | 5+ | 17.1 | 17.1 | 17.1 | |
| | | | | | |
| | Mean number of paying clients in the first month of sex work | 4.0 | 4.2 | 3.9 | 0.51 |
| Percent experienced physical violence in the past month | | 5.0 | 6.2 | 4.4 | 0.42 |
| Percent experienced coercion in the past month | | 4.6 | 2.1 | 6.0 | 0.07 |
| Percent experienced police violence in the past month | | 6.5 | 9.7 | 4.7 | 0.05 |
| Left Home/ living with parents | Left home | 61.8 | 57.3 | 64.3 | 0.16 |
| | Living with parents/others | 38.2 | 42.7 | 35.7 | |
| Age at start of sex work | < = 15 years | 19.1 | 13.5 | 22.4 | 0.02 |
| | 16–18 years | 43.2 | 41.2 | 44.3 | |
| | 19+ years | 37.7 | 45.3 | 33.3 | |
| | Mean age at start of sex work | 18.0 | 18.0 | 17.0 | 0.04 |

† There were 16 respondents who did not remember or answer the number of paying clients in the early period

**Table 5. Change in outcomes among young female sex workers who changed their primary typology of sex work between early and recent period of sex work compared to those who did not change, Mombasa, Kenya, 2015.**

| Outcomes | Overall Percent | Changed typology | Exp (B) | 95% CI for Exp (B) | | p |
| | | | | Lower | Upper | |
|---|---|---|---|---|---|---|
| Ever contacted by a peer/staff from NGO/CBO/FBO | 14.7 (N = 402) | No | 1.000 | | | |
| | | Yes | 0.962 | 0.534 | 1.734 | 0.898 |
| Ever tested for HIV | 77.4 (N = 372) | No | 1.000 | | | |
| | | Yes | 1.105 | 0.664 | 1.839 | 0.702 |
| Ever used a clinic run by NGO, CBO or FBO | 13.7 (N = 408) | No | 1.000 | | | |
| | | Yes | 0.883 | 0.493 | 1.581 | 0.676 |
| HIV prevalence | 7.3 (N = 317) | No | 1.000 | | | |
| | | Yes | 0.671 | 0.268 | 1.682 | 0.395 |

a clinic run by NGO/CBO/FBO. Further, 7.3% tested positive for HIV. The Odds Ratio and Confidence Intervals shows that there was no difference in prevalent HIV infection, programme reach, utilization of services among YFSW who changed their primary typology of sex work compared to those who did not change.

## Discussion

To the best of our knowledge, this is one of the first studies to focus on changes in sex work context, setting and practices within the early and recent period of sex work among YFSW in Kenya. Our findings show that even at the early period of sex work, there was a sub population of YFSW who were managed by third parties, had more than 5 paying clients, were meeting more than half of their living expenses through sex work and experienced police violence in the past month. Respondents reported an increase in self-management of their work, number of paying clients, meeting living expenses through the income of sex work and experiencing police violence by the recent period of sex work. In terms of typology, use of internet and mobile phone-based sex work increased from the early to the recent period of sex work. YFSW used more than one typology for sex work during both the early and recent period of sex work. The overlap in typology was high during both the periods of sex work with high overlap of entertainment venues among YFSW in all typologies of sex work. One third of YFSW changed their primary typology of sex work from the early to the recent period. Early sex work factors associated with change in primary typology were, street/ bus stop based sex work, experience of police violence and entry into sex work after 19 years. For YFSW, programme reach and service utilisation was low and prevalent HIV infection was moderately high at the early period of sex work. However, a change in primary typology by the respondents did not seem to change these outcomes among YFSW.

The ways in which sex workers operate continuously change, sometimes in response to contextual factors [17]. In a study conducted in Bangalore, India, changes related to age at initiation into sex work, type of partnerships, typology and setting of sex work and mobility of sex workers were noted over time [3]. Our study also shows that there were changes in sex work practices related to management, client volume, sex work income and police violence between the early and recent period of sex work. Self-management and autonomy impact a sex workers ability and freedom to negotiate condom use or regulate the number and type of clients [9] and higher sex work income is associated with consistent condom use with new clients among sex workers [41]. A systematic review conducted to understand the correlates of violence found evidence of the role of the work environment in shaping risks for violence among sex workers, with sex workers in street or public-place environments at highest risk of violence [42]. Another study conducted in India, also reported high violence in the first month of sex work [43]. While among YFSW, longer time in sex work may improve autonomy and income, however, the increase in the number of paying clients and police violence may also increase HIV related risks [44, 45]. It needs to be noted that even in the early period of sex work, a small sub population of YFSW respondents were managed by third parties, had high client volume and high violence. Programmes for YFSW in Mombasa, Kenya need to prioritise this sub-population during the early period of sex work.

Buzdugan and colleagues in their paper describe that the sex work industry is very fluid and new settings emerge over time [9]. Ward et al in their longitudinal study of 15 years with FSW in London found that almost a third of them had worked in all typologies of sex work [5]. Matolcsi et al also found that the breadth of the sex industry is large and individuals are often involved in more than one typology or setting over time or concurrently [46]. In our study a higher proportion of respondents reported using multiple typologies in a shorter span of time,

possibly a specific feature related to YFSW or Mombasa County. Entertainment venues were most popular among YFSW and would be an important site for interventions for programmes with YFSW in Mombasa. It is important to understand the interconnectedness between typologies and devise interventions at community and facility level that can maintain consistency in services across different typologies for all YFSW. Our findings also add to the body of emerging evidence that internet and cell phone-based sex work has increased globally [47]. Mahapatra et al found that use of cell phone among FSW was associated with increased HIV risk behaviors, independent of their place of solicitation [48]. With the increase in use of technology and cell phone in the lives of YFSW, interventions could devise technological and other solutions to reach individual YFSW on a regular basis irrespective of the typologies that they practice in [49].

In this study, we also found that a higher proportion of YFSW who changed their primary typology were street and bus stop based, experienced police violence in the past month and initiated sex work after 19 years of age at the early period of sex work. Programmes with YFSW should prioirtise reaching out to YFSWs with these characteristics with tailored interventions to provide support during the changes. Hendrickson et al also found in their study in Tanzania, that recently mobile FSW had a 25% increased risk of any recent experience of physical or sexual gender-based violence when compared with their non-mobile counterparts [50]. Though our study, in comparison, did not find any increased prevalent HIV incidence or decreased programme reach and service utilisation among YFSW who changed their primary typology of sex work, in general low programme reach and service utilisation and moderately high prevalent HIV incidence among the YFSW is very concerning and points towards the need to scale up HIV prevention programming with YFSW. Neufeld et al found that a large proportion of prevalent HIV infection among YFSW can be traced back to acquisition prior to entry into formal sex work. HIV prevention efforts targeting FSWs might overlook the opportunity to prevent HIV among the younger population if interventions fail to reach them early in their sexual life course [51].

Future research could focus on understanding risk and vulnerability linked to each typology to develop focused interventions to specifically target these typologies for YFSW. This information should also inform programmatic mapping and key population size estimation exercises to account for mobility of YFSW across sites [52]. Intervention models should be tested to explore feasibility of individualized outreach and service provision through use of technology. Future research should also focus on enhanced understanding of change in typology–who is changing, why are they changing, how does it impact programme reach etc.

## Limitations

The study examined the change in typology and its relationship with the sex work practices and contexts. First, we used the primary typology in the early and recent period to assess the change in typology. YFSWs might have changed their typologies between the period of early and recent periods and then shifted back to the early period typology in the recent period. However, we expect that these could be a small proportion, if any, considering the median duration in sex work was 2 years. Further, we present the different typologies of sex work that the respondents practiced in the early and recent periods separately, and show the overlap of typologies within and between periods. Secondly, the study retrospectively asked the sex work practices and contexts in the early period. There could be potential recall lapse, however, considering the short duration between early and recently period (median = 2 years) and since almost all of the respondents responded to the question, we consider such recall lapse is unlikely or minimum. Thirdly, the study sampled from the physical hotspots and hence may have left out respondents who seek clients and partners exclusively using the internet.

## Conclusions

YFSW change their sex work practices and typology of sex work within a short period of time. Understanding these changes in the early period of sex work could be important to design effective HIV prevention interventions to reach YFSW. Given global acknowledgement of the need to scale up interventions with YFSW, this paper provides new information that should be considered while designing and implementing interventions with YFSW.

## Supporting information

**S1 Table. List of questions.**
(DOCX)

**S2 Table. Change in primary typology of sex worker by duration of sex work among young female sex workers in Mombasa, Kenya 2015.**
(DOCX)

**S1 File. Inclusivity in global research.**
(DOCX)

## Acknowledgments

We thank the study participants and the research team involved in data collection in Mombasa. We thank the other members of the Transition Study Team including James Blanchard (University of Manitoba), Stephen Moses (University of Manitoba), Marie-Claude Boily (Imperial College London), Michael Pickles (Imperial College London), Daria Pavlova (Ukrainian Institute for Social Research After Oleksandr Yaremenko), Olga Balakireva (Ukrainian Institute for Social Research After Oleksandr Yaremenko), Larry Gelmon (University of Manitoba), Joshua Kimani (University of Manitoba), Fridah Muinde (National Syndemic Diseases Control Council), Martin Wafula Sirengo (National AIDS and STI Control Programme), Reynold Washington (India Health Action Trust), Sevgi Aral (Center for Disease Control), Lyle McKinnon (University of Manitoba), Yoav Keynan (University of Manitoba), Keith Fowke (University of Manitoba), Paul Sandstorm (University of Manitoba) and Eve Cheuk (University of Manitoba).

## Author Contributions

**Conceptualization:** Parinita Bhattacharjee, Shajy Isac, Marissa Becker, Sharmistha Mishra.

**Data curation:** Shajy Isac.

**Formal analysis:** Parinita Bhattacharjee, Shajy Isac, Huiting Ma.

**Funding acquisition:** Marissa Becker, Sharmistha Mishra.

**Methodology:** Shajy Isac, Marissa Becker, Sharmistha Mishra.

**Project administration:** Peter Gichangi, Marissa Becker.

**Resources:** Sharmistha Mishra.

**Supervision:** Helgar Musyoki, Peter Gichangi, Marissa Becker, Jan Hontelez, Sharmistha Mishra.

**Validation:** Jan Hontelez.

**Visualization:** Huiting Ma.

Writing – **original draft:** Parinita Bhattacharjee.

Writing – **review & editing:** Parinita Bhattacharjee, Shajy Isac, Helgar Musyoki, Peter Gichangi, Huiting Ma, Marissa Becker, Jan Hontelez, Sharmistha Mishra.

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
