## [Decision Letter · Decision Letter 0]

3 Apr 2023

PONE-D-22-24072Variability in the context between early and recent periods of sex work among young female sex workers in Mombasa, Kenya: a cross-sectional studyPLOS ONE

Dear Dr. Bhattacharjee,

Thank you for submitting your manuscript to PLOS ONE. After careful consideration, we feel that it has merit but does not fully meet PLOS ONE’s publication criteria as it currently stands. Therefore, we invite you to submit a revised version of the manuscript that addresses the points raised during the review process.

The second reviewer points out several methodologic issues that must be addressed to support this work and lead to publication.==============================

We look forward to receiving your revised manuscript.

Kind regards,

Jill Blumenthal

Academic Editor

PLOS ONE

Journal Requirements:

2. Please include a complete copy of PLOS’ questionnaire on inclusivity in global research in your revised manuscript. Our policy for research in this area aims to improve transparency in the reporting of research performed outside of researchers’ own country or community. The policy applies to researchers who have travelled to a different country to conduct research, research with Indigenous populations or their lands, and research on cultural artefacts. The questionnaire can also be requested at the journal’s discretion for any other submissions, even if these conditions are not met.  Please find more information on the policy and a link to download a blank copy of the questionnaire here: https://journals.plos.org/plosone/s/best-practices-in-research-reporting. Please upload a completed version of your questionnaire as Supporting Information when you resubmit your manuscript.”

3. You indicated that you had ethical approval for your study. Please clarify whether minors (participants under the age of 18 years) were included in this study. If yes, in your Methods section, please ensure you have also stated whether you obtained consent from parents or guardians of the minors included in the study or whether the research ethics committee or IRB specifically waived the need for their consent.

“MB, SM were supported by CIHR grants - MOP 13044 and FDN13455. PB was supported by BMGF grant - INV006613. The funders had no role in study design, data collection and analysis, decision to publish or preparation of the manuscript.”

“We thank the study participants and the research team involved in data collection in Mombasa. The study was funded by an operating grant (MOP-13044) from the Canadian Institutes of Health Research (CIHR) and analyses funded via CIHR grant FDN 13455. Analysis for this manuscript was also supported by the Bill & Melinda Gates Foundation (BMGF) under grant INV-006613. The views expressed herein are those of the authors and do not necessarily reflect the official policy or position of the funders. “

“MB, SM were supported by CIHR grants - MOP 13044 and FDN13455. PB was supported by BMGF grant - INV006613. The funders had no role in study design, data collection and analysis, decision to publish or preparation of the manuscript.”

6. One of the noted authors is a group or consortium [The Transitions Team ]. In addition to naming the author group, please list the individual authors and affiliations within this group in the acknowledgments section of your manuscript. Please also indicate clearly a lead author for this group along with a contact email address.

7**. **Please include your tables as part of your main manuscript and remove the individual files. Please note that supplementary tables (should remain/ be uploaded) as separate "supporting information" files

Additional Editor Comments (if provided):

The second reviewer points out several methodologic issues that need to be addressed to support this work and lead to publication.

Reviewers' comments:

Reviewer's Responses to Questions

**Comments to the Author**

1. Is the manuscript technically sound, and do the data support the conclusions?

Reviewer #1: No

Reviewer #2: No

2. Has the statistical analysis been performed appropriately and rigorously? 

Reviewer #1: No

Reviewer #2: No

3. Have the authors made all data underlying the findings in their manuscript fully available?

Reviewer #1: Yes

Reviewer #2: Yes

4. Is the manuscript presented in an intelligible fashion and written in standard English?

Reviewer #1: No

Reviewer #2: No

5. Review Comments to the Author

Reviewer #1: In the manuscript “Variability in the context between early and recent periods of sex work among young female sex workers in Mombasa, Kenya: a cross-sectional study,” authors use data from a 2015 cross-sectional survey with 408 female sex workers to assess differences in context and practices of sex work between the first month of the woman’s engagement in sex work and more recent sex work because differences may implicate a need for different time-period tailored HIV prevention programs. While the research question is important and study has potential for high impact, it has several shortcomings which need to be addressed.

Abstract: Consider briefly defining ‘typology’ and ‘early’ vs. ‘recent’ as these terms are not readily apparent/familiar to readers who do not commonly conduct research/programs with this population. The Results statement beginning with “higher proportion among street/bus stop typology” seems incomplete and should be reworded for clarity.

Intro: For clarity, consider defining “early” vs “recent” period and age criteria for “YFSW” early on in the intro (i.e., at first reference). Also, consider expanding on the hypothesis that HIV/STI risk may vary by change in typology and sex work practices over time. Why and how might change in typology and sex practices change HIV risk? Why/how this could impact HIV prevention strategies? This will further justify research question and need for the study.

Methods:

- As a key aim of the study is to assess change in typology of sex work over time, how adequate were current recruitment methods in comprehensively identifying/recruiting participants across dimensions of typology? i.e., if participants were recruited from sex work venues, would participant who engaged in home-based or internet-based sex work be adequately identified? The period of “early” vs. “recent” could have different impact based on how long the participant had been involved with sex work. For example, we might expect little variation in someone who had only been employed for 2 months vs 24 months in sex work. How was this handled?

- Were participants reimbursed for participation? What was the length of the survey? How was the privacy and safety of the participants ensured?

- More details regarding how the predictors “physical violence,” “experience of coercion,” and “living condition” were assessed and analyzed is needed.

- The study outcomes should be identified and defined consistently. For example, lines 151-152 describe planned assessment of the impact of primary typology on “HIV prevalence” and “program reach.” Is prevalence the right outcome? Perhaps HIV status (negative vs. positive) is more appropriate? How is program reach defined and measured? The Statistical Analysis section only discusses testing the association between change in primary typology and factors associated with change in typology. This is discrepant from the prior section.

Results:

- Table 1 should depict the spread of values for individuals engaged in sex work <2 years.

- Table 2 should include physical violence and living condition as described in the Methods

- Table 3 text should describe overlap between typologies. How is this handled/depicted in the table?

- A key is needed to explain shading in Figure 1

- Table 4 and 5 don’t seem to be hypothesis driven or aligned with the paper. For example, the analysis presented answers what factors drove change in typology and impact of change in typology on outcome. I think the real question that authors are interested in is the switch from a lower-risk to a higher-risk typology and this is not being assessed in the current analysis.

Overall comments: The manuscript could benefit from grammar/spelling proofing.

Reviewer #2: Thank you for the opportunity to review this interesting study. The authors aim to assess changes in the context and typology of sex work across early and recent phases of sex work and implications for HIV programmes. Strengths include the peer-led nature of data collection and the potential significance of addressing this more nuanced aspect of sex work, which is a gap in the literature. However, there are a number of substantial methodologic issues and questions that I would recommend be addressed to support rigor and clarity of this work and its potential contribution to the literature. Below I've identified some point-by-point questions and comments for the authors to consider. Additionally, there are considerable language and grammatical issues that require addressing for the manuscript to be fully intelligible and clear.

Abstract & Title

1. The relevance of this work is not quite clear based on the current framing of the title and introduction.

2. More clarity regarding which programmes are being referred to and why they would only be focusing on 'several years' after entry into sex work is unclear from the abstract.

3. The title and objective would benefit from more unpacking and clarity, especially as the study relates to HIV or other health outcomes.

4. The conclusions provided in the abstract are fairly vague and would benefit from more specific recommendations and clarity.

Introduction

5. The introduction would be strengthened by revising the opening paragraph to provide more nuance and clarity regarding the broader study context and justification - why do typologies of sex work and changes over time matter, and how could this information be used in HIV or other health programming would ideally be clearer much earlier in the paragraph.

6. Some awkward terms are used (eg, scanty) that should be addressed, and much of the description of concepts (eg, which 'programmes' are being referred to in the first sentence? what is meant by 'critical outcomes' in the objective??) is vague, leaving the reader wondering exactly what is meant. More precise language and clarification of key concepts would strengthen the introduction and all sections of the manuscript. The hypotheses for the study also require clarification.

Methods

7. The sole reliance on descriptive methods is a substantial methodologic weakness. I appreciate the descriptive analysis, but wonder why no odds or risk ratios (bivariate) were provided? ORs/RRs and 95% CIs could provide more nuanced effect estimates and interpretation over sole reliance on p-values and comparing percentages. Additionally, a more focused analysis that includes multivariable modeling to adjust for confounding may provide a stronger study design and weight of evidence.

Results

8. The authors may consider referring to pimps/managers as 'third parties' as this is a more neutral term that describes these types of sex industry roles, that may help avoid the stigma, myths and misconceptions that are often attributed to social norms surrounding 'pimps' and their role in sex transactions.

9. There is a large amount of descriptive data provided, and inclusion of more bivariate effect sizes and 95% CIs, as well as some multivariable modeling addressing key hypotheses and relationships between variables, could provide a stronger set of results.

6. PLOS authors have the option to publish the peer review history of their article (what does this mean?). If published, this will include your full peer review and any attached files.

Reviewer #1: No

Reviewer #2: No

---

## [Author Response · Author response to Decision Letter 0]

15 Jun 2023

Journal Requirements:

Thank you for sharing the links. We have now ensured that the manuscript meets PLOS ONE’s style requirements including file names

2. Please include a complete copy of PLOS’ questionnaire on inclusivity in global research in your revised manuscript. Our policy for research in this area aims to improve transparency in the reporting of research performed outside of researchers’ own country or community. The policy applies to researchers who have travelled to a different country to conduct research, research with Indigenous populations or their lands, and research on cultural artefacts. The questionnaire can also be requested at the journal’s discretion for any other submissions, even if these conditions are not met. Please find more information on the policy and a link to download a blank copy of the questionnaire here: https://journals.plos.org/plosone/s/best-practices-in-research-reporting. Please upload a completed version of your questionnaire as Supporting Information when you resubmit your manuscript.”

We have now included the PLOS’ questionnaire on inclusivity in global research as supporting information (S1File). We have also included a subsection ‘Inclusivity in global research’ to your Methods section (page 11, Line 213-214) 

3. You indicated that you had ethical approval for your study. Please clarify whether minors (participants under the age of 18 years) were included in this study. If yes, in your Methods section, please ensure you have also stated whether you obtained consent from parents or guardians of the minors included in the study or whether the research ethics committee or IRB specifically waived the need for their consent.

The participants for the study included adolescent girls and young women from 14 to 24 years congregating at sex work hotspots. The local IRB waived the need for parental consent. We have now included the same in the Methods section (page 7 Line 134-137).

“MB, SM were supported by CIHR grants - MOP 13044 and FDN13455. PB was supported by BMGF grant - INV006613. The funders had no role in study design, data collection and analysis, decision to publish or preparation of the manuscript.”

We have revised the funding statement in the manuscript and has also included the same in our cover letter

“We thank the study participants and the research team involved in data collection in Mombasa. The study was funded by an operating grant (MOP-13044) from the Canadian Institutes of Health Research (CIHR) and analyses funded via CIHR grant FDN 13455. Analysis for this manuscript was also supported by the Bill & Melinda Gates Foundation (BMGF) under grant INV-006613. The views expressed herein are those of the authors and do not necessarily reflect the official policy or position of the funders. “

“MB, SM were supported by CIHR grants - MOP 13044 and FDN13455. PB was supported by BMGF grant - INV006613. The funders had no role in study design, data collection and analysis, decision to publish or preparation of the manuscript.”

We have now revised the acknowledgement section (page 21, lines 427- 458) and the funding statement (cover letter).

6. One of the noted authors is a group or consortium [The Transitions Team]. In addition to naming the author group, please list the individual authors and affiliations within this group in the acknowledgments section of your manuscript. Please also indicate clearly a lead author for this group along with a contact email address.

We have now added names of other Transitions Team members and their affiliations in the acknowledgement (page 21, lines 427-458). We have also added the name of the lead author and the contact email (page 2, lines 29-30).

7. Please include your tables as part of your main manuscript and remove the individual files. Please note that supplementary tables (should remain/ be uploaded) as separate "supporting information" files

We have added the tables as part of the main manuscript (pages 26-29) and have added the supplementary table as separate supporting information files (S1 Table, S2 Table and S1 File).

Additional Editor Comments (if provided):

The second reviewer points out several methodologic issues that need to be addressed to support this work and lead to publication.

We appreciate the comments of the second reviewer. We have made revisions as best as we could. We have, in particular, based on reviewers’ feedback, added Odds Ration and 95% confidence interval along with p values in Table 5 (page 29) to provide nuanced analysis. However this is a descriptive paper and the analysis is used to describe the changes in sex work practices, context, typologies and primary typology of sex work during the early and recent periods of sex work, in Mombasa, Kenya and therefore the scope of the paper is not to predict risk. This descriptive analysis has been done to inform programme planning and implementation. We acknowledge the limitations of a descriptive analysis and have made revisions as best as we can to address the concerns of the second reviewer. However, we contend that a descriptive analysis also provides an important set of new findings, as recently noted by Platt and Lesko et al. We have cited this framework for descriptive epidemiology in our revised manuscript.

Reviewers' comments:

Reviewer #1: In the manuscript “Variability in the context between early and recent periods of sex work among young female sex workers in Mombasa, Kenya: a cross-sectional study,” authors use data from a 2015 cross-sectional survey with 408 female sex workers to assess differences in context and practices of sex work between the first month of the woman’s engagement in sex work and more recent sex work because differences may implicate a need for different time-period tailored HIV prevention programs. While the research question is important and study has potential for high impact, it has several shortcomings which need to be addressed.

1. Abstract: Consider briefly defining ‘typology’ and ‘early’ vs. ‘recent’ as these terms are not readily apparent/familiar to readers who do not commonly conduct research/programs with this population. The Results statement beginning with “higher proportion among street/bus stop typology” seems incomplete and should be reworded for clarity.

Thank you for the feedback. In the revised manuscript we have defined the terms typology, early and recent more clearly in the abstract and have also revised the result statement (page 3, lines 42, 48).

2. Intro: For clarity, consider defining “early” vs “recent” period and age criteria for “YFSW” early on in the intro (i.e., at first reference). Also, consider expanding on the hypothesis that HIV/STI risk may vary by change in typology and sex work practices over time. Why and how might change in typology and sex practices change HIV risk? Why/how this could impact HIV prevention strategies? This will further justify research question and need for the study.

Thank you very much for the comments. In the introduction, we have defined the key words at the first reference. We have also added new evidence to show the variation in HIV and STI risk by sex work typology and why change in typology may facilitate HIV risk among FSW. We hope that this will justify the need to understand change in typology and sex work practices for improved prevention programming (pages 5-6, lines 72-115). 

2. Methods:

- As a key aim of the study is to assess change in typology of sex work over time, how adequate were current recruitment methods in comprehensively identifying/recruiting participants across dimensions of typology? i.e., if participants were recruited from sex work venues, would participant who engaged in home-based or internet-based sex work be adequately identified? The period of “early” vs. “recent” could have different impact based on how long the participant had been involved with sex work. For example, we might expect little variation in someone who had only been employed for 2 months vs 24 months in sex work. How was this handled?

- Were participants reimbursed for participation? What was the length of the survey? How was the privacy and safety of the participants ensured?

Thank you for the comment. The study started with mapping of physical locations where female sex workers congregated to solicit clients and/or where sex work-related sexual activities took place. The method and the results of the mapping exercise is described in a paper written by our Transitions study team member. We have also added this reference in the method section (page 8, line 140). The recruitment method for this study sampled only physical hotspots identified during the mapping exercise which included local brew dens, night clubs, sex den, bars with and without lodging, brothels, streets, public spaces, guesthouses, restaurant, massage parlours, truck stops, video dens and homes. Hence, the sampling approach adopted for the study represented all physical sites, including homes. However, the study did not sample virtual sites and would have left out respondents exclusively using internet. We have now included this exclusion as a limitation of the study (Page 19, lines 404-405). 

We agree that the period of “early” vs. “recent” could have different impact on change in primary typology based on how long the participant had been involved with sex work. However, we did not find a variation by duration in sex work and change in typology as defined in years (even when we considered less than 6 months when it was less than 1 year). Therefore, we presented this in years in the table and did not break into many categories considering the sample size in each cells. We did however, as suggested, conduct a sensitivity analysis. The change in primary typology by years in sex work (in months for below 1 year, 1-2 years, 3-4 years and above 5 years is presented now as supporting information in the paper (S2 Table). 

The participants were reimbursed Ksh 850 (USD 10) for their time. The actual survey took 1hour and 30 minutes including the collection of biological samples. The survey took place in private spaces that were identified by the study team. The spaces were located within the Drop-in Centres and clinics managed by our local partner, International Centre for Reproductive Health – Kenya (ICRH-K) or government clinics identified by the peer educators as safe and friendly spaces. All the rooms where the face-to-face interviews and biological sample collections took place had privacy for the respondents. All respondents were given a unique ID to protect confidentiality and this ID was attached to their questionnaire and biological sample. Hard copies of the filled questionnaires and other documents were stored in locked cabinets, accessible only by members of the research team. The electronic data was entered in a password protected computer and stored in a password protected server. Only the research team had access to the data (page 8, lines 150-159). 

- More details regarding how the predictors “physical violence,” “experience of coercion,” and “living condition” were assessed and analyzed is needed.

Thank you so much for your comment. We have now added a supporting information table sharing relevant questions that were used to define the variables (S1 Table). This will help readers understand how the variables used in the analysis were defined.

- The study outcomes should be identified and defined consistently. For example, lines 151-152 describe planned assessment of the impact of primary typology on “HIV prevalence” and “program reach.” Is prevalence the right outcome? Perhaps HIV status (negative vs. positive) is more appropriate? How is program reach defined and measured? The Statistical Analysis section only discusses testing the association between change in primary typology and factors associated with change in typology. This is discrepant from the prior section.

Thank you so much for the comment. We have now renamed the HIV prevalence outcome as prevalent HIV infection starting from introduction section (page 7, line 124) and then used consistently throughout the rest of the manuscript. We have now defined programme reach and service utilisation in the methods section (page 10-11, lines 189 -193) and also revised the statistical analysis section to include the outcomes (page 10, 206-209).

3. Results:

- Table 1 should depict the spread of values for individuals engaged in sex work <2 years.

We have now revised Table 1 to show the spread of values for respondents engaged in sex work < 2 years (page 26).

- Table 2 should include physical violence and living condition as described in the Methods

Thank you for your comment. We have added supplementary information describing the questions used to define these variables (S1 Table).

- Table 3 text should describe overlap between typologies. How is this handled/depicted in the table?

Thank you for your comment. We apologise for the confusion created. Table 3 describes the distribution in typology in the early and recent period of sex work among young female sex workers. We have revised the title of the table (page 27) for greater clarity. The table does not describe overlap. Overlap is described in Figure 1.

- A key is needed to explain shading in Figure 1

Thank you for the comment. The key has been added to Figure 1 now.

- Table 4 and 5 don’t seem to be hypothesis driven or aligned with the paper. For example, the analysis presented answers what factors drove change in typology and impact of change in typology on outcome. I think the real question that authors are interested in is the switch from a lower-risk to a higher-risk typology and this is not being assessed in the current analysis.

Thank you for your question. We have clarified that our study cannot answer the question surrounding whether switching from a lower risk to higher risk typology has occurred. Rather, we aimed to describe if differences in primary typology were associated with prevalent HIV infection (Table 5) (page 29). For Table 4 (page 28), our focus was solely on describing the characteristics of those who reported a change in their primary typology, and not testing a hypothesis. We have revised the manuscript to clarify the descriptive nature of the study (page 10, lines 198 -203).

4. Overall comments: The manuscript could benefit from grammar/spelling proofing.

Thank you for the comment. We have done a grammar and spelling proof of the manuscript.

Reviewer #2: Thank you for the opportunity to review this interesting study. The authors aim to assess changes in the context and typology of sex work across early and recent phases of sex work and implications for HIV programmes. Strengths include the peer-led nature of data collection and the potential significance of addressing this more nuanced aspect of sex work, which is a gap in the literature. However, there are a number of substantial methodologic issues and questions that I would recommend be addressed to support rigor and clarity of this work and its potential contribution to the literature. Below I've identified some point-by-point questions and comments for the authors to consider. Additionally, there are considerable language and grammatical issues that require addressing for the manuscript to be fully intelligible and clear.

5. Abstract & Title

5.1 The relevance of this work is not quite clear based on the current framing of the title and introduction.

Thank you for the feedback. We have changed the title (page 1, lines 1-2) and the abstract to clearly communicate the objective of the study and its relevance.

5.2. More clarity regarding which programmes are being referred to and why they would only be focusing on 'several years' after entry into sex work is unclear from the abstract.

Thank you for the feedback. We have clarified that we are referring to HIV prevention programmes with female sex workers when we refer to ‘programmes’. The HIV prevention programmes with female sex workers reach female sex workers several years after they self-identify as sex workers. Considering prevalent HIV incidence among young female sex workers, we feel that there is need to understand the context and setting in the early years of sex work to design HIV prevention programmes with young female sex workers to ensure early reach and access.

. 

5.3. The title and objective would benefit from more unpacking and clarity, especially as the study relates to HIV or other health outcomes.

Thank you for your feedback. We have revised the title to now relate more with the study objectives (page 1, lines 1-2). The study relates to HIV related outcomes rather than broader health outcomes. We have clarified this point throughout the paper.

5.4. The conclusions provided in the abstract are fairly vague and would benefit from more specific recommendations and clarity.

Thank you for the feedback. We have revised the conclusions to provide clear recommendations based on the study findings (page 4, lines 66-68)

6. Introduction

6.1. The introduction would be strengthened by revising the opening paragraph to provide more nuance and clarity regarding the broader study context and justification - why do typologies of sex work and changes over time matter, and how could this information be used in HIV or other health programming would ideally be clearer much earlier in the paragraph.

6.2 Some awkward terms are used (eg, scanty) that should be addressed, and much of the description of concepts (eg, which 'programmes' are being referred to in the first sentence? what is meant by 'critical outcomes' in the objective??) is vague, leaving the reader wondering exactly what is meant. More precise language and clarification of key concepts would strengthen the introduction and all sections of the manuscript. The hypotheses for the study also require clarification.

The introduction section has been edited extensively to incorporate the reviewers feedback and suggestions (pages 5-6, lines 72-115).

7. Methods

7.1 The sole reliance on descriptive methods is a substantial methodologic weakness. I appreciate the descriptive analysis, but wonder why no odds or risk ratios (bivariate) were provided? ORs/RRs and 95% CIs could provide more nuanced effect estimates and interpretation over sole reliance on p-values and comparing percentages. Additionally, a more focused analysis that includes multivariable modeling to adjust for confounding may provide a stronger study design and weight of evidence.

We are thankful for your feedback. In particular, based on the reviewers’ feedback, we have now added Odds Ratio and 95% confidence interval along with p values in Table 5 (page 30). The scope of the paper is to look at patterns rather than predict risk and hence we used descriptive analysis. This descriptive analysis has been done to inform programme planning and implementation. We acknowledge the limitations of a descriptive analysis and have made revisions as best as we can to address the concerns of the second reviewer. However, we contend that a descriptive analysis also provides an important set of new findings, as recently noted by Platt and Lesko et al. We have cited this framework for descriptive epidemiology in our revised manuscript (page 10, lines 198- 200).

8.Results

8.1 The authors may consider referring to pimps/managers as 'third parties' as this is a more neutral term that describes these types of sex industry roles, that may help avoid the stigma, myths and misconceptions that are often attributed to social norms surrounding 'pimps' and their role in sex transactions.

Thank you so much for the feedback. In the paper narrative we have replaced pimps/ managers as third parties (page 16, line 315 and page 17, line 346).

8.2. There is a large amount of descriptive data provided, and inclusion of more bivariate effect sizes and 95% CIs, as well as some multivariable modeling addressing key hypotheses and relationships between variables, could provide a stronger set of results.

Thank you for this comment. Based on your feedback we have added Odds Ratio and 95% Confidence Interval in Table 5 (page 29). However, we would like to affirm that this is a descriptive paper and the analysis is used to describe the changes in sex work practices, typologies and primary typology of sex work during the early and recent periods of sex work, in Mombasa Kenya. We then characterize those young female sex workers who change their primary typology, and assess the impact of this change on critical outcomes. This descriptive analysis has been done to inform program planning and implementation. We note that you appreciate the descriptive analysis.

---

## [Editor Report · Decision Letter 1]

5 Jul 2023

Changes in context, typology and programme outcomes between early and recent periods of sex work among young female sex workers in Mombasa, Kenya: a cross-sectional study

PONE-D-22-24072R1

Dear Dr. Bhattacharjee,

We’re pleased to inform you that your manuscript has been judged scientifically suitable for publication and will be formally accepted for publication once it meets all outstanding technical requirements.

Kind regards,

Jill Blumenthal

Academic Editor

PLOS ONE

Additional Editor Comments (optional):

We appreciate your attention to the responses offered and have no further recommended edits.

---

## [Editor Report · Acceptance letter]

17 Jul 2023

PONE-D-22-24072R1 

Changes in context, typology and programme outcomes between early and recent periods of sex work among young female sex workers in Mombasa, Kenya: a cross-sectional study 

Dear Dr. Bhattacharjee:

I'm pleased to inform you that your manuscript has been deemed suitable for publication in PLOS ONE. Congratulations! Your manuscript is now with our production department. 

Kind regards, 

on behalf of

Dr. Jill Blumenthal 

Academic Editor

PLOS ONE